# iTRAQ-Based Phosphoproteomic Analysis Exposes Molecular Changes in the Small Intestinal Epithelia of Cats after *Toxoplasma gondii* Infection

**DOI:** 10.3390/ani13223537

**Published:** 2023-11-16

**Authors:** Bintao Zhai, Yu-Meng Meng, Shi-Chen Xie, Jun-Jie Peng, Yang Liu, Yanhua Qiu, Lu Wang, Jiyu Zhang, Jun-Jun He

**Affiliations:** 1Key Laboratory of Veterinary Pharmaceutical Development, Lanzhou Institute of Husbandry and Pharma-Ceutical Sciences, Chinese Academy of Agricultural Sciences, Ministry of Agriculture and Rural Affairs, Lanzhou 730050, China; zhaibintao@163.com (B.Z.); qyhranglin@163.com (Y.Q.); 2State Key Laboratory of Veterinary Etiological Biology, Key Laboratory of Veterinary Parasitology of Gansu Province, Lanzhou Veterinary Research Institute, Chinese Academy of Agricultural Sciences, Xujiaping 1, Lanzhou 730046, China; 13109339679@163.com (Y.-M.M.); pengjunjielz@163.com (J.-J.P.); 3College of Veterinary Medicine, Shanxi Agricultural University, Taigu, Jinzhong 030801, China; xieshichen221@163.com (S.-C.X.); llyhtw1105@163.com (L.W.); 4College of Life Science, Ningxia University, Yinchuan 750021, China; liuyangnihao@139.com; 5Key Laboratory of Veterinary Public Health of Yunnan Province, College of Veterinary Medicine, Yunnan Agricultural University, Kunming 650201, China

**Keywords:** phosphoproteomics, *Toxoplasma gondii*, post-translational modification, cat

## Abstract

**Simple Summary:**

*Toxoplasma gondii* is a parasite that invades nucleated cells and causes changes in the cellular signal transduction network. This study aimed to perform a comprehensive analysis of the phosphorylated proteome in the small intestine cells of the definitive host following *T. gondii* infection. We used titanium dioxide affinity chromatography to enrich phosphopeptides in cat small intestinal epithelia infected with *T. gondii* and then used iTRAQ technology to quantify the phosphopeptides. A total of 4998 phosphopeptides, 3497 phosphorylation sites, and 1805 phosphoproteins were identified. Among the differentially expressed phosphoproteins, 68 were down-regulated and 637 were up-regulated. The bioinformatics analysis revealed that these phosphoproteins were involved in various cellular processes such as actin cytoskeleton reorganization, cell necroptosis, and the MHC immune process. The study confirmed that *T. gondii* infection leads to extensive changes in the phosphorylation of cat intestinal proteins. This is the first study to report global protein phosphorylation alterations in feline small intestinal epithelia following *T. gondii* infection. These findings provide a theoretical basis for understanding the interaction between *T. gondii* and its definitive host, which could have implications for the development of treatments or preventive measures.

**Abstract:**

*Toxoplasma gondii*, an obligate intracellular parasite, has the ability to invade and proliferate within most nucleated cells. The invasion and destruction of host cells by *T. gondii* lead to significant changes in the cellular signal transduction network. One important post-translational modification (PTM) of proteins is phosphorylation/dephosphorylation, which plays a crucial role in cell signal transmission. In this study, we aimed to investigate how *T. gondii* regulates signal transduction in definitive host cells. We employed titanium dioxide (TiO_2_) affinity chromatography to enrich phosphopeptides in the small intestinal epithelia of cats at 10 days post-infection with the *T. gondii* Prugniuad (Pru) strain and quantified them using iTRAQ technology. A total of 4998 phosphopeptides, 3497 phosphorylation sites, and 1805 phosphoproteins were identified. Among the 705 differentially expressed phosphoproteins (DEPs), 68 were down-regulated and 637 were up-regulated. The bioinformatics analysis revealed that the DE phosphoproteins were involved in various cellular processes, including actin cytoskeleton reorganization, cell necroptosis, and MHC immune processes. Our findings confirm that *T. gondii* infection leads to extensive changes in the phosphorylation of proteins in the cat intestinal epithelial cells. The results of this study provide a theoretical foundation for understanding the interaction between *T. gondii* and its definitive host.

## 1. Introduction

*Toxoplasma gondii* is an obligate intracellular protozoan parasite that can infect most warm-blooded animals for a long time, or even for life [1]. Although *T. gondii* infection is largely asymptomatic for immunocompetent individuals, it is life-threatening for individuals with weakened immunity or those who are immunocompromised, such as pregnant or AIDS patients [2]. It is well known that only members of the feline family act as strict definitive hosts for *T. gondii*, including domestic cat and wild cat, and sexual reproduction occurs exclusively in the small intestine epithelium of cats [3,4]. An infected cat can shed millions of oocysts that serve as the main reservoir of *T. gondii*. Therefore, cats play a vital role in the transmission of *T. gondii*. It may not only directly threaten human health but may also contribute to greater environmental pollution, causing food-borne pathogens [5]. However, there are few studies on *T. gondii* infection in cats, especially regarding the biological changes in the small intestine and oocysts in cats after infection.

Protein phosphorylation is a widespread and essential post-translational modification (PTM) that plays a crucial role in pathogen infection by regulating protein activity [6]. Previous research has shown that phosphorylated proteins are involved in the regulation of numerous cellular signals with host cells, influencing nearly one-third of protein functions [7]. It is evident that protein phosphorylation is critical for controlling the interaction between pathogens and host cells [8]. *T. gondii*, for instance, has evolved to reprogram host biological processes by modifying the phosphorylation status of host proteins [9]. Notably, several virulence factors of *T. gondii*, such as ROP18, ROP16, and ROP17, function as protein kinases [10,11]. During the invasion of host cells, *T. gondii* releases ROP18 via small vesicles derived from apical secretory organelles called rhoptries. These vesicles enable ROP18 to bind to immune-related GTPases (IRGs) on the outer surface of the parasite-containing vacuole, thereby phosphorylating them [12]. This phosphorylation prevents the clearance of IRGs within inflammatory monocytes and IFN-γ-activated macrophages, contributing to parasite survival in vivo and promoting virulence. Similarly, ROP17 targets members of the IRG family and this interaction complements the action of ROP18 [13,14]. In *T. gondii* strains of type I or III, ROP16 functions as a transcription regulator of host cells and disrupts host functions by phosphorylating the stat protein [15]. This leads to the modulation of the STAT3/6 signaling pathway and subsequently affects the downstream host cytokine IL-12. Additionally, ROP16 induces the phosphorylation and nuclear translocation of STAT5, resulting in the production of protective immunity [16]. Studies utilizing these virulence proteins or gene knock-out strains have elucidated various virulence mechanisms employed by *T. gondii* to regulate the host’s biological and immune responses. However, utilizing the wild-type strain can unveil the overall influence of *T. gondii* infection on host cells. For instance, upon infecting host cells, *T. gondii* can induce the phosphorylation of the host cell protein Bcl-2-associated death promoter (BAD), subsequently inhibiting host cell apoptosis by regulating the PI3K-PKB/AKT signaling pathway [17]. The global impact of *T. gondii* infection on the phosphoproteome of human foreskin fibroblast cells has been studied, revealing that *T. gondii* modifies host actin and the plasma membrane by altering the host phosphorylation status [18]. Considering the significant role of cats in the dissemination of *T. gondii*, investigating the phosphoproteome of the small intestine of cats becomes crucial in unraveling the sexual replication mechanism and how *T. gondii* reprograms the biological processes of its definitive host.

With the continuous improvement of the quantitative protein analysis by LC-MS/MS, the protein phosphorylation analysis was determined as more rapid and sensitive [19]. In this study, the quantitative phosphorylated proteomics method based on iTRAQ combined with TiO_2_ affinity chromatography was used to analyze the phosphorylation status of the intestinal protein of cats during *T. gondii* oocysts’ discharging period. Our findings provide new resources for elucidating the interaction between *T. gondii* and its definitive host cells.

## 2. Materials and Methods

### 2.1. Animals, Parasite Infection, and Sample Collection

Six female Chinese Li Hua breed domestic cats, aged 7–9 months, were obtained from local breeders and housed in a controlled environment. Before the commencement of the experiment, all cats underwent serological testing using an enzyme-linked immunosorbent assay (ELISA) kit to detect major feline viruses, including feline immunodeficiency virus, feline leukemia virus, feline calicivirus, and feline parvovirus [20]. Additionally, a modified agglutination test (MAT) was performed to confirm that the cats were seronegative for specific anti-*T. gondii* antibodies, with a cut-off of 1:25 [21]. During the first three weeks of the experiment, the cats were provided with commercial diets (Royal Canine Inc., St. Charles, MO, USA) and had access to drinking water ad libitum to minimize any potential influence of diet on the experimental outcomes. Throughout the experiment, the cats were fed once a day according to their energy requirements, with drinking water available as desired.

The *T. gondii* Prugniuad (Pru) strain is capable of forming cysts in the mouse brain and oocysts in the small intestinal epithelium of cats, making it an ideal candidate for experimental infection in cats. The *T. gondii* Pru strain was maintained via passage in Kunming mice. The number of *T. gondii* cysts was adjusted to 100 cysts/mL of phosphate-buffered saline (PBS) with a pH of 7.4, as determined via microscopic observation. Six cats were randomly assigned to either the experimental group or the control group, with three cats in each group. Each cat in the experimental group was infected via intragastric inoculation with 100 cysts/mL of sterile PBS (2 mL per cat). The control group received the same amount of PBS. Ten days post-infection, we took steps to collect tissue samples from the cats. First, all cats were euthanized, and the small intestinal epithelia cells of the cats were cleansed twice with PBS, a crucial step to ensure the removal of all intestinal contents. Finally, using sterile techniques, we gently scratched the intestinal epithelial tissue of the cats. A small portion of the collected samples was used for DNA extraction and identification, while the remaining samples were quickly frozen in liquid nitrogen after weighing and then transported to BGI-shenzhen for sequencing using dry ice transport. Proper protocols were followed to ensure the harmless treatment of all remaining experimental animal tissues and animal facilities.

### 2.2. Confirming T. gondii Infection via PCR Analysis

Genomic DNA was extracted from each harvested tissues using the TIANamp Genomic DNA Kit (TianGenTM, Beijing, China, Lot: U8812) in accordance with the manufacturer’s instructions. A *T. gondii B1* gene PCR assay was employed to detect *T. gondii* infection in the cat intestinal epithelial tissues. The primers and protocol used in this study were consistent with the previously described methods [22]. The PCR products were electrophoresed on a 3% Regular-Agarose-G10 gel stained with a GoldView Nucleic acid gel stain (Yeasen Biotechnology Co., Ltd., Shanghai, China, Lot: A2A1634), and the banding pattern was visualized using UV transillumination (BIO-RAD, CA, USA, Serial No: 76S/07933). All electrophoresis PCR products were subjected to a positive and negative control.

### 2.3. Protein Extraction, Quality Control, Reductive Alkylation, and Digestion

In brief, the protein extraction, digestion, and iTRAQ labeling were performed following the previously described methods [23,24]. The intestinal sample was ground into powder and suspended with protein lysis buffer 3 (7 M Urea, 1 mM PMSF, 2 mM EDTA, 20 mM Tris HCl, and 10 mM DTT at pH 8.5), supplemented with phosphatase inhibitor (PhosSTOP, Roche, Shanghai, China). The suspension was then subjected to ultrasonic treatment on ice for 5 min. The supernatant was collected via centrifugation at 25,000× *g* for 20 min at 4 °C. An amount of 10 mM dithiothreitol (DTT, 56 °C, 1 h) was added to the supernatant for deoxidation. After recovery at room temperature, 55 mM IAM was added for alkylation for 45 min (in the dark). Protein concentration was determined using the Bradford protein quantification method [25]. SDS-PAGE treatment was performed and each protein sample (300 μg) was digested overnight (4 h and 8 h) at 37 °C using Trypsin Gold (Promega, Madison, WI, USA Lot: V5280) at a ratio of 1:40. The collected mixture was desalted using a Strata X solid phase extraction column (Phenomenex, Torrance, CA, USA). Finally, the peptide sample was vacuumed and stored at −80 °C until further use.

### 2.4. Enrichment of Phosphorylated Peptides and iTRAQ Labeling

We used Titanium dioxide (TiO_2_, GL Sciences, Tokyo, Japan) to enrich the labeled phosphorylated peptides (Peptide sample: TiO_2_ = 1:4) following the previously described methods [23]. The mixture was suspended with a 1 mL loading buffer and shaken for 10 min at room temperature. Next, the iTRAQ labeled reagent was added and shaken on the rotator at 37 °C for 1 h. The iTRAQ labeling was performed according to the manufacturer’s instructions and the peptide was labeled using the 8-plex iTRAQ labeling kit (Applied Biosystems/MDS Sciex, Foster City, CA, USA). The samples collected post *T. gondii* infection were labeled with reagents 115, 116, and 117, while the control samples were labeled with reagents 118, 119, and 121, respectively. The mixture was precipitated with 12,000 g for 5 min and the supernatant was discarded. The pellets were rinsed with 2 mL of wash buffer, centrifuged at 12,000× *g* for 30 s, and the supernatant was discarded. This washing step was repeated four times. The last pellets were suspended in 600 μL of elution buffer 1 and shaken for 20 min, followed by centrifugation at 12,000× *g* for 30 s to collect the supernatant. Then, the 500 μL elution buffer 2 was added, shaken for 20 min, and centrifuged at 12,000× *g* for 30 s to collect the supernatant. Finally, the collected supernatant was mixed and dried using vacuum centrifugation, and the resulting samples were analyzed using LC-MS/MS.

### 2.5. LC-MS/MS Analysis and Peptide Identification

UHPLC (UltiMate 3000, Thermo, Milan, Italy) combined with tandem mass spectrometer Q-Exactive HF (Thermo Fisher Scientific, San Jose, CA, USA) was used to analyze the phosphopeptides separated by LC-MS/MS. The extracted peptide sample was re-dissolved with 300 μL 0.1% trifluoroacetic acid (TFA), a high PH reversed-phase (RP) column (Thermo Scientific Pierce, CA, USA) was used for the separation and grouping of the peptides according to the manufacturer’s instructions, and the sample was then freeze-dried. The sample was redissolved with 20 μL buffer A (2% ACN, 0.1% FA), centrifuged at 20,000 rpm, and the supernatant was collected and then separated using UHPLC. The sample was first enriched and desalted in a trap column and then connected in series with a self-contained C18 column (75 μm inner diameter, 3 μm grain size, and 25 cm column length). The samples were eluted from the column using a linear gradient of solution buffer B (98% ACN, 0.1% FA), running at a flow rate of 300 nL/min. The elution gradient was as follows: 0–5 min, 5% buffer B; 5–45 min, 5–25% buffer B; 45–50 min, 25–35% buffer B; 50–52 min, 35–80% buffer B; 52–54 min, 80% buffer B; and 54–60 min, down to 5% buffer B. The end of the nanoliter liquid phase separation was directly connected to the mass spectrometer. The separated peptides in the liquid phase were ionized using a nano-electrospray ionization (nanoESI) source and injected into the Q-Exactive HF tandem mass spectrometer for data-dependent acquisition (DDA) mode detection. The main parameters of mass spectrometry are set as follows: the first-order mass spectrum was obtained at a resolution of 120,000 in the scanning range of 350-1500 *m*/*z*. The automatic gain control (AGC) target and maximum injection time of the primary mass spectrometer are 1 × 10^6^ and 30 ms, respectively. The spectrum of the secondary mass spectrometry was obtained using the following parameters: resolution = 30,000; starting *m*/*z* was fixed at 100; maximum injection time was 30 ms; and AGC target was 5 × 10^4^. The mode of ion fragmentation was performed using high energy collision dissociation (HCD), with a normalized collision energy of 30 eV.

### 2.6. Database Search and Bioinformatics Analysis

The database (uniprot_felis_catus_and_toxoplasma_gondii_20190108.fasta) search was performed using Proteome Discoverer 1.4 (Xcalibur, Thermo Fisher Scientific) with the Mascot search engine (version 2.3.02, Matrix Science, http://www.matrixscience.com) following the previously described operational steps [26,27]. The specific operating parameters used were as follows: trypsin was designated as the cleavage enzyme, allowing for up to two missing cleavages. The mass tolerances for the peptide and fragment was set at 20 ppm and 0.05 Da, respectively. The fixed modifications included Carbamidomethyl (C), iTRAQ8plex (N-tern), iTRAQ8plex (K). Variable modifications considered were Oxidation (M), Acetyl (Protein N-term), Deamidated (NQ), Phosphorylation (S/T/Y), and iTRAQ8plex (Y). For the identification and quantification of phosphopeptides, the screening criteria were set as *p* ≤ 0.05, with the selection of phosphopeptides and phosphosites on proteins based on a phosphoRS probability ≥ 0.75. The threshold for the differential expression of phosphopeptides between the experimental and control groups was set at a ratio ≥ 1.5 or ≤0.667 and *p* ≤ 0.05.

After the Gene Ontology (GO) method (http://www.geneontology.org, accessed on 9 October 2023) annotation, the biological domain of the identified DEPs can be categorized into three aspects: molecular function (MF), cellular component (CC), and biological process (BP). To classify and functionally annotate the DEPs, the Cluster of Orthologous Groups of proteins (COG) database was utilized. The pathway analysis of the DEPs was conducted using the Kyoto Encyclopedia of genes and genome (KEGG, https://www.genome.jp/kegg/pathway.html, accessed on 30 October 2023) database. The KEGG enrichment analysis was performed with preset parameters and the results were visualized using enrichment bubble plots. The STRING database (http://string-db.org/, accessed on 24 August 2023) was searched for a protein–protein interaction (PPI) network; in our study, the STRING database was employed to identify the interaction relationship and intensity between DEPs, as well as to examine the upstream and downstream connections between a single phosphopeptide and other associated phosphopeptides. Subsequently, the PPI network of DEPs’ interaction was screened based on combined scores (≥0.7, high confidence) and represented as lines and circles using Cytoscape software (Version 3.9.1, https://apps.cytoscape.org/, accessed on 25 June 2022). Additionally, the plug-in of minimal common oncology data elements (MCODE) was employed to analyze the highly interconnected clusters within the PPI network.

## 3. Results

### 3.1. Small Intestinal Epithelial Cells of Cat Infected with Toxoplasma gondii

After infecting the cat with *T. gondii* at 10 DPI, the DNA was extracted from the intestinal epithelium and amplified using PCR. The results of gel electrophoresis showed that the experimental group was positive for the *B1* gene at 10 DPI and there was no positive result for the control group (Figure 1). This demonstrates that the cat intestinal epithelium infection model was successfully established, which laid a foundation for further research.

### 3.2. Phosphopeptide Landscape and Identification of Differential Phosphopeptides

We profiled a phosphoproteomic profiling to investigate the changes in the intestinal epithelia of cats infected with the *T. gondii* Pru strain. In our study, iTRAQ-phosphorylated proteomics technology was used to qualitatively and quantitatively analyze the alterations in phosphorylated protein levels following *T. gondii* infection in cats. In this study, we obtained 247,246 spectra. Under the filtering condition of false positive FDR ≤ 0.01 at the spectral level, a total of 34,038 spectra, 4998 phosphopeptides (Appendix A), and 3497 phosphorylation sites (phosphoRS probability ≥ 0.75) were identified on 1805 phosphorylated proteins. The length distribution of the phosphopeptides was primarily concentrated between 10 to 35 amino acids (Figure 2A), with the majority consisting of polypeptides modified by a single phosphorylation site (91.88%) (Figure 2B). Notably, we observed that SRRM2, SRRM1, LIMA1, CTNND1, OSBP, LMO7, NPM1, PLEC, NUCKS1, LMNA, BCLAF1, TNKS1BP1, LARP1, MAP4, MAP1B, FAM83H, and HDAC2 had more than 20 phosphorylation sites. The distribution of phosphorylated amino acid residues in the identified phosphorylation sites is illustrated in Figure 2C, with 3112 (88.99%) phosphorylated at serine (pSer), 349 (9.98%) at threonine (pThr), and 36 (1.03%) at tyrosine (pTyr). Figure 2D depicts the distribution of phosphorylation sites with each phosphorylated protein, with one phosphorylation site accounting for the majority (1090, 60.39%). The repeatability analysis, conducted using the coefficient of variation (CV), reveals that 0.44 of the total phosphoprotein is identified when the CV is 20% (Figure 3). Following conditional screening (ratio ≥ 1.5 or ≤0.667, and *p* ≤ 0.05), we identified 705 differentially phosphorylated proteins in this study. Among these, 637 were up-regulated, while 68 were down-regulated (Figure 4).

### 3.3. COG Classification Analysis and GO Annotation Analysis of DEPs

To further investigate the functional significance of the differentially expressed phosphoproteins (DEPs) identified from the small intestine of cats infected with *T. gondii*, we conducted a COG classification analysis and a gene ontology (GO) annotation analysis. The DEPs were categorized into 23 functions based on the COG function analysis (Figure 5A). The results revealed that the most abundant DEPs were involved in translation, ribosome structure and biogenesis, and signal translation mechanisms. In the GO annotation analysis, the phosphoproteins were annotated and classified into molecular function (MF), cellular component (CC), and biological process (BP) (Figure 5B,C). The MF analysis showed that both up- and down-regulated phosphoproteins were associated with protein binding and ATP binding. In the CC category, both up-regulated and down-regulated phosphoproteins were found in the cytoplasm, nucleus, and cytoplasm. Interestingly, in the BP category, the up-regulated phosphoprotein was involved in a positive regulation of transcription from the RNA polymerase II promoter and a negative regulation of transcription from the RNA polymerase II promoter, while the down-regulated phosphoprotein was involved in proteolysis and the regulation of the apoptotic process.

### 3.4. Enrichment Analysis of Differentially Expressed Phosphoproteins

Firstly, we performed a functional enrichment analysis based on GO to determine the potential functional significance of the 705 differentially phosphorylated proteins. The results revealed significant enrichment of these phosphorylated proteins in 320 CC terms, 389 MF terms, and 1994 BP terms (Appendix A). Based on the *p*-value, the top 10 enriched GO terms for each category are shown in Figure 6A. Among the CC categories, the top three were non-membrane-bounded organelle, intracellular non-membrane-bounded organelle, and actin cytoskeleton. In the BP category, the top three terms were actin filament organization, protein complex subunit organization, and negative regulation of cell proliferation. The top three enriched terms in the MF category were nucleic acid binding, phosphoglucomutase activity, and helicase activity. To gain further insight into the pathways involved in *T. gondii*-infected cats, we conducted a KEGG enrichment analysis. A total of 705 differential phosphoproteins were annotated into the KEGG database, involving 279 pathways (Appendix A). Notably, several important pathways (*p*-value ≤ 0.05) were significantly enriched among these phosphoproteins, including the Wnt signaling pathway and the NOD-like receptor signaling pathway (Figure 6B).

### 3.5. Protein–Protein Interactions (PPI)

The PPI network was constructed to gain insight into the regulatory mechanisms of phosphorylation modification and determine the interactions between phosphoproteins. A total of 168 differential phosphoproteins (up: 158, down: 10) were mapped into the protein network database (Figure 7). The MCODE analysis revealed relevant subnetworks from six MCODE clusters in Figure 7 (Ribosome biogenesis, spliceosome, Senine/threonine-protein kinase, Vial carcinogenesis, RNA transport, and Antigen processing and presentation), as shown in Appendix A. These analysis results are consistent with the GO enrichment results.

## 4. Discussion

*Toxoplasma gondii* is a wide-spreading intracellular parasite that regulates the phosphorylation levels of multiple proteins in the host cell, thereby influencing host metabolic and immune response pathways [28]. Protein phosphorylation is a common regulatory mechanism in organisms, with over 30% of cellular proteins being phosphorylated. This process plays a crucial role in cell signal transduction, the regulation of cell proliferation, development, differentiation, and apoptosis [29]. Although there have been few reports on the phosphorylation modification of *T. gondii* following host cells’ infection, understanding the phosphorylation process of *T. gondii* is vital in comprehending its ability to evade host immunity [30,31]. Therefore, the objective of this study was to investigate the phosphorylation process of the small intestine cells in the definitive host after *T. gondii* infection and identify differential molecules. This research will provide a theoretical foundation for the prevention and treatment of toxoplasmosis.

Multiple studies have shown that infection with different strains of *T. gondii* can have varying effects on host cell phosphorylation [32,33,34]. For instance, infection with type I strain (BK) has been found to suppress the tyrosine phosphorylation of STAT1 in host cells. On the other hand, infection with type I strain RH, type II strain PRU, and type III strain CL14 can induce the degradation of phosphorylated STAT1 in the nucleus of host cells. *T. gondii* type I (BK) can suppress the tyrosine phosphorylation of STAT1 in host cells, while infection with type I (RH), type II (PRU), and type III (CL14) can induce the degradation of phosphorylated STAT1 in the nucleus of host cells [35]. Moreover, in laboratory experiments using THP-1 macrophages, *T. gondii* infection has been shown to trigger the phosphorylation of PKB/Akt at Ser473 and Thr308. This activation of PKB/Akt leads to the phosphorylation of the bad factor (one of the pro-apoptotic BH3-only proteins) at Ser112, effectively inhibiting host cell apoptosis [17]. Although these studies have provided some insights into phosphorylation events during the interaction between *T. gondii* and host cells, a comprehensive investigation of phosphorylation has not yet been conducted.

To address this gap in knowledge, our study aimed to perform a comprehensive analysis of the phosphorylated proteome in the small intestine cells of the definitive host following *T. gondii* infection. We employed an iTRAQ-based MS method combined with TiO_2_ affinity chromatography to investigate the phosphorylation process.

For eukaryotes, protein phosphorylation sites typically occur in serine/threonine/tyrosine residues, with varying biological processes and proportions involved [36]. However, serine phosphorylation accounts for 80-90% of the total phosphorylation sites [6,37]. In our study, we observed that the majority of protein phosphorylation sites occurred on serine residues (88.99%), followed by threonine (9.98%) and tyrosine (1.03%), after *T. gondii* infection in cats (Figure 2C). These findings align with previous studies, which have shown a similar distribution of protein phosphorylation sites in oocysts and tachyzoites of the *T. gondii* PYS strain [38]. Our results suggest that serine residues are the main target for protein phosphorylation following *T. gondii* infection in cats. In this study, SRRM2, SRRM1, LIMA1, CTNND1, OSBP, LMO7, NPM1, PLEC, NUCKS1, LMNA, BCLAF1, TNKS1BP1, LARP1, MAP4, MAP1B, FAM83H, and HDAC2 are considered as highly phosphorylated proteins with over 20 phosphorylation sites. Among them, MAP4 and MAP1B are microtubule-associated proteins (MAPs), and phosphorylation of MAPs is considered a crucial regulatory factor for microtubule stability. Phosphorylation reduces the affinity of MAPs, leading to their detachment from microtubules and subsequent instability [39]. These findings indicate that *T. gondii* infection induces extensive changes in host cell proteins at the phosphorylation level.

All the differentially expressed phosphoproteins that were identified were assigned to their corresponding entries in the GO and COG databases. Among the entries corresponding to MF, protein binding and ATP binding were observed (Figure 5). This indicates that energy metabolism is closely linked to the proliferation of *T. gondii* in the host cells. In the CC category, the majority of DEPs were enriched in the cytosol, nucleus, and cytoplasm. This suggests that *T. gondii* regulates biological processes occurring in the cytosol, nucleus, and cytoplasm via the modification of the phosphorylation status in phosphoproteins.

This study identifies a significant number of up-regulated phosphoproteins involved in the regulation of the apoptotic process. Apoptosis plays a crucial role in regulating the host’s response to parasite infection [40,41]. Increased apoptosis may contribute to the spread of intracellular pathogens [42] or induce immunosuppression [43]. Conversely, inhibiting cell apoptosis can prevent the spontaneous apoptosis of host cells and the elimination of *T. gondii* and the apoptotic host by phagocytes. At the same time, it allows the parasite to evade the cytotoxic effects mediated by T cells and NK cells, thereby altering the immune response of the host. This immune evasion mechanism of *T. gondii* has been previously demonstrated in experiments [44], where infected host cells showed resistance to pro-apoptotic stimuli. The loss of the NF-κB subunit p65 in *T. gondii*-infected cells results in the loss of their anti-apoptotic effects. Furthermore, the release of cytochrome C from mitochondria in infected host cells is inhibited, and the invasion of *T. gondii* is facilitated via the migration of p50 and p65 subunits to the nucleus. Experiments have demonstrated that *T. gondii* infection can activate and phosphorylate PKB/Akt [17,45]. The activated PKB/Akt phosphorylates the pro-apoptotic factor, Bad factor, and the phosphorylated Bad factor inhibits Bax from the cytoplasm to the shift of mitochondria, which inhibits the occurrence of apoptosis events mediated by mitochondria. Interestingly, the KEGG results indicated a significant enrichment of DEPs in necroptosis, an emerging form of programmed cell death that serves as a new automatic defense mechanism when apoptosis fails (Figure 6B). A total of 11 DEPs were enriched in this pathway. Hsp90, a highly-conserved molecular chaperone, plays a role in protein and various kinase stabilizations [46]. The formation of the Hsp90 complex is necessary for the development of necrotic bodies following the activation of RIP3 and MLKL and the stimulation of TNF [47]. Furthermore, the inhibition of HSP90 has been shown to prevent necroptosis induced by TNF [48]. Thus, targeting the necroptosis pathway in the host using HSP inhibitors could serve as a potential avenue for investigating the interaction between *T. gondii* and its definitive hosts.

In the GO enrichment analysis of the significantly changed phosphoprotein, the GO terms “actin cytoskeleton” and “actin filament organization” were found to be significantly enriched (Figure 6A). This suggests that the infection of *T. gondii* is closely linked to the reorganization of the host cytoskeleton. Unlike other pathogens, *T. gondii* does not enter the host cell via endocytosis, and the host cell plays a relatively passive role in the invasion process. Instead, *T. gondii* forms a parasitophorous vacuole during invasion, making it crucial to maintain the integrity of the actin cytoskeleton. Previous studies by Silva et al. demonstrated that rupturing the actin cytoskeleton of host cells using cytochalasin D and microfilament depolymerization agents led to a reduction in *T. gondii* tachyzoite invasion [49]. Furthermore, the *T. gondii* protein kinase IRE1 has been found to regulate cytoskeleton remodeling and cell migration via its interaction with actin cross-linking factor fibrin a, enabling the parasite to spread within the host [50]. In this study, it was observed that certain proteins are involved in multiple pathways related to cytoskeletal regulation, such as PKC and Zyxin. PKC is a calcium ion-activated serine/threonine protein kinase, which can finally activate the NF-κB cell signaling pathway via phosphorylation of the IKK complex [51,52]. In this study, a significant increase in the phosphorylation level of PKC was detected, proving that the effect of the NF-κB pathway activated by it has been enhanced [53]. Zyxin, on the other hand, exhibited seven up-regulated phosphorylation sites. Zyxin is an essential component of the focal adhesion protein and functions as a messenger in cytoskeleton-related cell signal transduction [54]. It is responsible for transmitting focal adhesion-related stimuli to the corresponding genes and regulating actin reorganization in the cytoskeleton. This leads to the reorganization of the actin bundle [55,56].

Notably, in this study, several immune-related pathways were highly enriched, including endocytosis, necroptosis, and the nod-like receptor signaling pathway (Figure 6B). Among them, there were changes in the expression of phosphoproteins associated with the major histocompatibility complex (MHC). For example, FLA-I and FLAI-H were up-regulated. These genes are considered classic loci in cats and they play a crucial role in studying the response of epitope-specific antiviral drugs CD8+ T cells in cats [57]. MHC class I proteins and endogenous antigen peptides (such as certain *T. gondii* proteins that can be degraded by the proteasome) combine to form a peptide-MHC I complex in the endoplasmic reticulum. Subsequently, this complex is presented on the cell surface to activate the corresponding CD8+ T cells and convert them into sensitized cytotoxic T lymphocytes (CTLs), which exert a cytotoxic effect to eliminate target cells [58]. Numerous studies have demonstrated the importance of the CTL response in both the development of *T. gondii* infection and immunity against *T. gondii* [59,60,61]. In this study, the up-regulation of phosphoproteins in the MHC I pathway suggests that the host has undergone a CTL immune response to restrict the proliferation of *T. gondii*.

Once *T. gondii* infects host cells, it initiates a fast and persistent phosphorylation of STAT3. The parasite utilizes host STAT3 to impede the production of IL-12 and TNF-α, which are typically triggered by LPS, and to inhibit the production of pro-inflammatory factors [62]. Among the up-regulated transcription factors, NF-κB1 (p50/p105) occupies an important position. According to the KEGG analysis, p50 is one of the key transcription factors in the NF-κB signaling pathway. The NF-κB pathway serves as a central regulator of inflammation, immunity, and stress response by controlling the gene expression of cytokines, transcription factors, and stress responses [63,64]. In this study, the NF-κB signaling pathway was significantly enriched and up-regulated. These findings suggest that *T. gondii* infection in the small intestine might affect the host’s immune response to *T. gondii* by up-regulating this signaling pathway. Prior studies have also found that the DNA binding capacity of p50 is regulated by the phosphorylation of residue Ser 337, which influences both the positive and negative control of NF-κB transcription [65].

## 5. Conclusions

In this study, we delineate the global phosphorylated proteome of feline intestinal proteins infected with *T. gondii*. Through the integration of the iTRAQ and the TiO_2_ affinity chromatography analysis, we identified 3497 phosphorylation sites within 1805 phosphoproteins. These significantly regulated phosphoproteins were further classified using the GO analysis and the KEGG enrichment analysis. Our analysis revealed that the phosphoproteins altered due to *T. gondii* infection were dispersed across various cellular compartments and were associated with an array of biological functions, including actin cytoskeleton reorganization and necroptosis. Phosphoprotein appears to play a role in cats’ immune response to parasites. This study provides a new valuable resource for further research on the functional changes in the cat’s small intestine during *T. gondii* infection.

## Figures and Tables

**Figure 1 animals-13-03537-f001:**
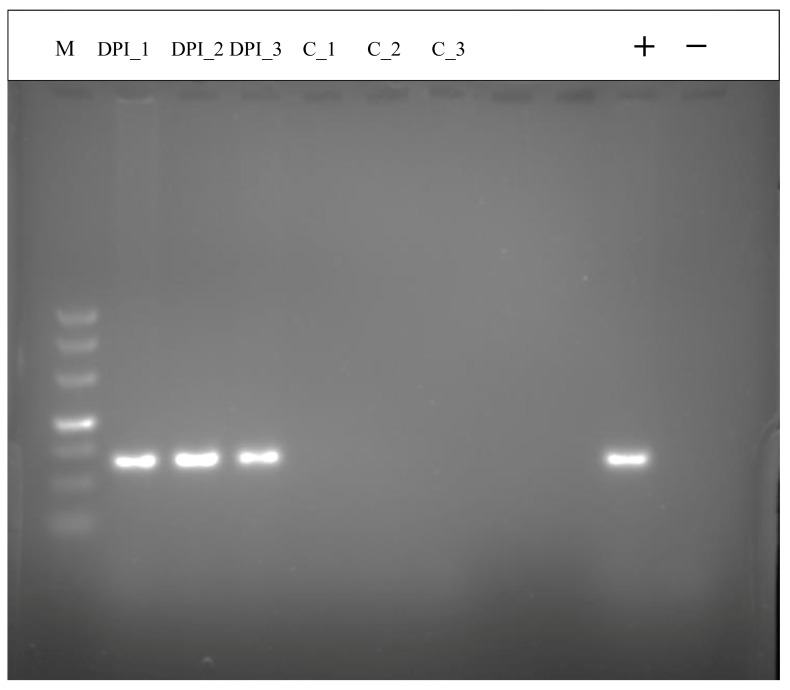
The results of PCR for the samples. Identification of *T. gondii* infected cat small intestine samples at 10 DPI by *T. gondii B1* gene. The order of the sample holes is M: Takara DNA marker (500 bp); E-group (lanes 2–4, 10 DPI_1, 10 DPI_2, 10 DPI_3), C-group (lanes 5–7, C_1, C_2, C_3), and P: *T. gondii* Pru strain PCR positive control; N: PCR negative control, E-group: experimental group, C-group: control group.

**Figure 2 animals-13-03537-f002:**
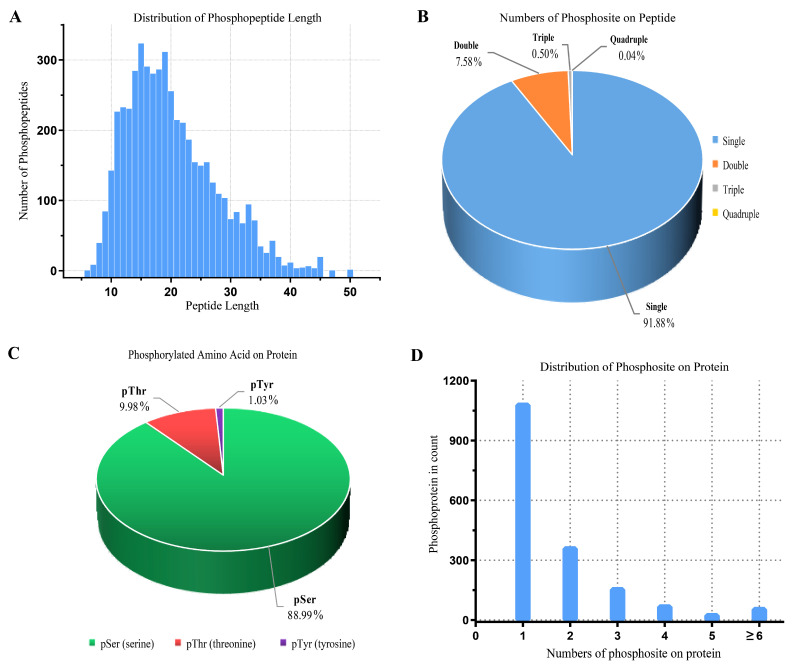
The properties of phosphopeptides of cat small intestine cells infected by *Toxoplasma gondii*. (**A**) The length distribution of phosphorylated peptides. (**B**) Polyphosphorylation distribution of phosphorylated peptides. (**C**) Distribution of phosphorylation on serine (pSer), threonine (pThr), and tyrosine (pTyr) for all phosphorylation sites. (**D**) Distribution of 1805 phosphoproteins based on identification of single or multiple phosphosites per protein.

**Figure 3 animals-13-03537-f003:**
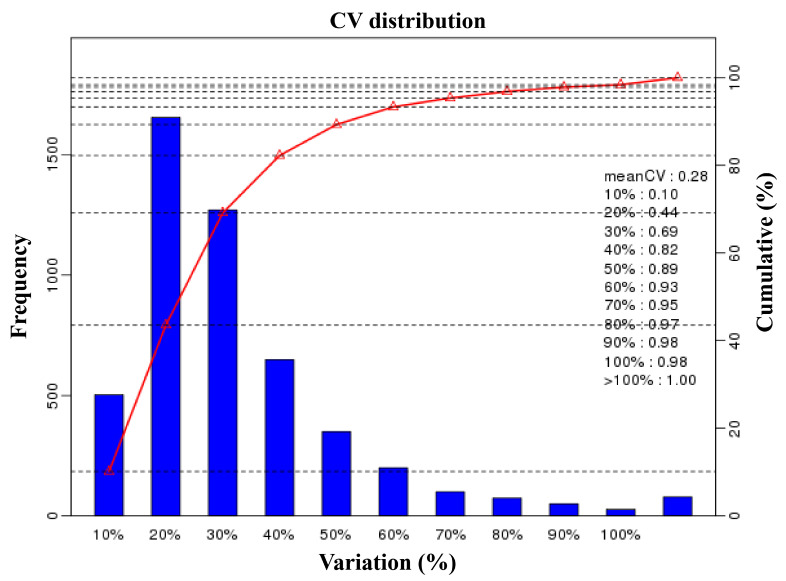
The repeatability evaluation of phosphopeptide quantification based on the coefficient of variation (CV). The percentage on *x*-axis represents the values of CV. The left *y*-axis indicates the number of phosphopeptides and the right *y*-axis indicates the cumulative percentage of phosphopeptides.

**Figure 4 animals-13-03537-f004:**
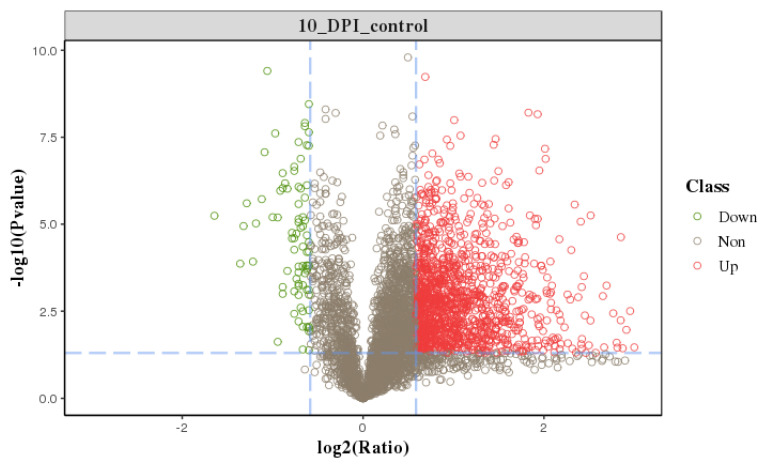
Volcano map of the differential phosphoproteins. Significant DEPs are shown as a red (up) or green (down) dot. No significant difference between the expressions of phosphoproteins is shown as a brown dot.

**Figure 5 animals-13-03537-f005:**
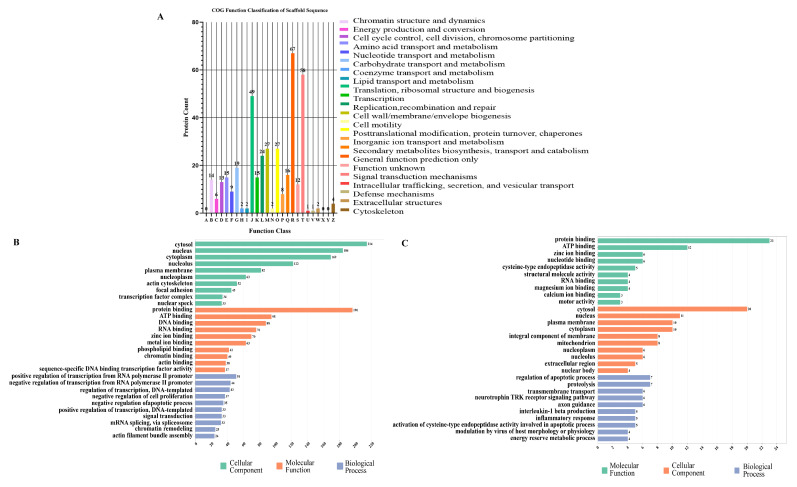
Functional classification of phosphoproteins in feline intestine infected by *Toxoplasma gondii*. (**A**) Functional annotation of the DEPs based on COG. (**B**) GO functional classification of the up-regulated phosphoproteins based on gene ontology. (**C**) GO functional classification of the down-regulated phosphoproteins based on gene ontology.

**Figure 6 animals-13-03537-f006:**
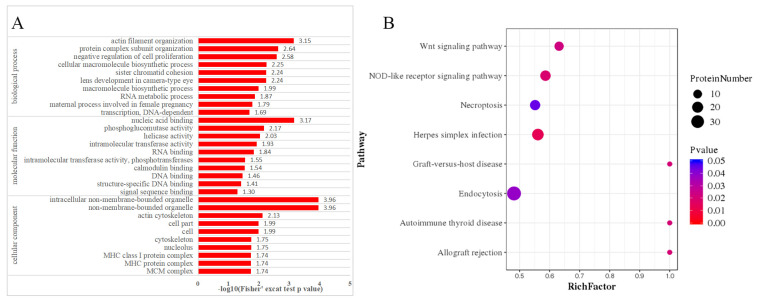
GO enrichment analysis and KEGG pathway analysis of the differentially expressed phosphoproteins of feline small intestine infected with *Toxoplasma gondii*. (**A**) The 30 most enriched GO terms belonging to biological process, molecular function, and cellular component are shown. (**B**) The pathway obtained from KEGG pathway enrichment analysis. Rich factor represents the ratio of the DEP’s number and the number of proteins annotated in the pathway. The higher the rich factor is, the higher the degree of enrichment.

**Figure 7 animals-13-03537-f007:**
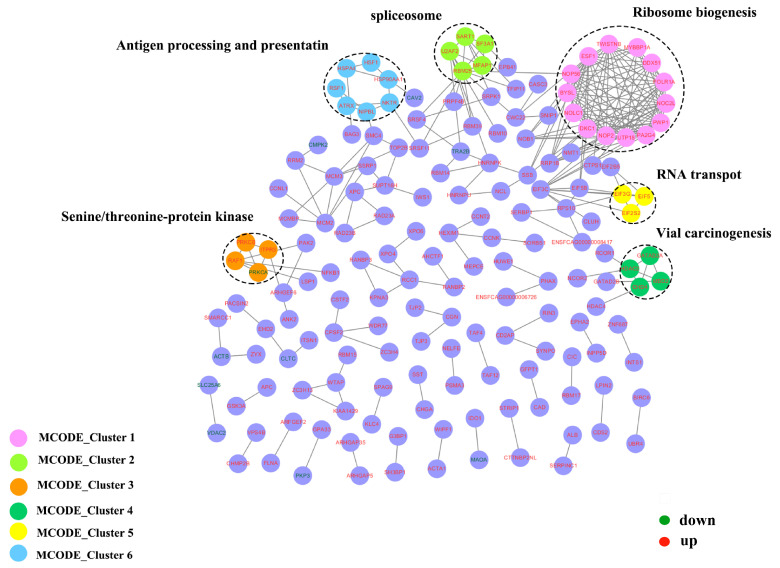
Protein–protein interaction (PPI) networks for the differential phosphoproteins. Protein interaction network of the differential phosphoproteins of feline small intestine. Nodes represent the phosphoproteins and edges represent interactors between phosphoproteins. Protein names are in red and green fonts, red indicates upregulation, and green indicates downregulation. The dotted circles represent six MCODE clusters (MCODE clusters 1: Ribosome biogenesis, MCODE clusters 2: spliceosome, MCODE clusters 3: Senine/threonine-protein kinase, MCODE clusters 4: Vial carcinogenesis, MCODE clusters 5: RNA transport, and MCODE clusters 6: Antigen processing and presentation).

## Data Availability

The mass spectrometry data have been submitted to the iProX database with the project ID IPX0002163000 or at Mendeley data https://data.mendeley.com/datasets/rgn4fm9cr7/1.

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
