# Peer review of "iTRAQ-Based Phosphoproteomic Analysis Exposes Molecular Changes in the Small Intestinal Epithelia of Cats after Toxoplasma gondii Infection"

_animals, 2023, doi:10.3390/ani13223537_

Round 1
Reviewer 1 Report
Comments and Suggestions for Authors
Major comments:
I had the pleasure to review animals-2632951 which investigated the global phosphorylated proteome of feline intestinal proteins infected with T. gondii, a common parasitic infection in cats. Using integrated proteomic analysis, the researchers identified significant changes in the phosphorylation of certain proteins, which were associated with various cellular processes and biological functions. The manuscript is generally well written and results are correctly described. However, the methods require more details and some formats need to be modified. The manuscript is of value to the field and furthers our understanding of T. gondii infection.
Minor comments:
Below are some suggestions·for improvement:
Point·1. line4 and line 25, titled “Toxo-plasma gondii” should be·“ Toxoplasma gondii ”, ' - 'should be removed.
Point·2. Lines·26, and 27. “The researchers” can be changed·to "We", "they then...,"can be changed to “and then...”.
Point 3. Line 60. “Patient” should be plural.
Point·4. line·171, and·lines·245-250. the paragraph has·format problems, it need·to·be modified.
Point·5. “3.1. Small intestinal epithelial model of cat infected with Toxoplasma gondii” should be revised
Point·6. lines·145, 255, 262. gene name “B1”·should be italic.
Point·7. Lines·159, 160, 164, “℃” font should be Palatino Linotype.
Point·8. In Figure 1, please provide the size of B1 gene and each lane
Point 9. Please revise line 285“revealed that 0.44 of the total……”
Piont 10. The characters of Figure 5 are hard to read. Please provide font size that can be read.
Point 11. Line 403. “80%-90%” there should be a space between “ - ”.
Piont 12. Lines 410, 410.·English Punctuations “,” should be used between phonologically proteins.
Point 13. The references·format contain certain issues that require rectification, notably in references 2-15, and 43, among others. In addition, some proper nouns need italics, such as line 665, “in vivo”.
Comments on the Quality of English Languagewell
Author Response
9 Nov, 2023
Dr. Ewelina Bik
Assistant Editor for
Animals
Dear Dr. Ewelina Bik,
Re: Manuscript ID: animals-2632951.R1
On behalf of all co-authors, I would like to thank you and the reviewers very much for favorable comments and constructive suggestions on our manuscript (MS) animals-2632951. The reviewers considered our MS to be of general interest to the readership of Animals, and recommended the acceptance of our MS for publication after revisions.
Therefore, we have revised the MS strictly according to the reviewers’ comments and suggestions. We used the “tracked changes” mode in the WORD to show the revised/changed text in the revised MS. Two MS files are uploaded: the “clean version” as “manuscript”, and the one showing “tracked changes” as “supplementary material”. In the following, we detail our point-by-point responses to the comments and suggestions of Reviewer #1. We made all our responses in blue colour for clarity.
Responses to the comments and suggestions of Reviewer #1
Comments and Suggestions for Authors
Major comments:I had the pleasure to review animals-2632951 which investigated the global phosphorylated proteome of feline intestinal proteins infected with T. gondii, a common parasitic infection in cats. Using integrated proteomic analysis, the researchers identified significant changes in the phosphorylation of certain proteins, which were associated with various cellular processes and biological functions. The manuscript is generally well written and results are correctly described. However, the methods require more details and some formats need to be modified. The manuscript is of value to the field and furthers our understanding of T. gondii infection.Responses: We thank Reviewer #1 very much for favorable comments on our manuscript (MS).
Minor comments:
Below are some suggestions·for improvements:
Point·1. line4 and line 25, titled “Toxo-plasma gondii” should be·“ Toxoplasma gondii ”, ' - 'should be removed.
Responses: we thank the Reviewer 1# very much and we have revised this accordingly.
Point·2. Lines·26, and 27. “The researchers” can be changed·to "We", "they then...,"can be changed to “and then...”.
Responses: Thank you very much for your constructive suggestions. We have revised this accordingly.
Point 3. Line 60. “Patient” should be plural.
Responses: revised accordingly.
Point·4. line·171, and·lines·245-250. the paragraph has·format problems, it need·to·be modified.
Responses: revised accordingly.
Point·5. “3.1. Small intestinal epithelial model of cat infected with Toxoplasma gondii” should be revised
Responses: revised accordingly. (line 237)
Point·6. lines·145, 255, 262. gene name “B1”·should be italic.
Responses: revised accordingly.
Point·7. Lines·159, 160, 164, “℃” font should be Palatino Linotype.
Responses: revised accordingly.
Point·8. In Figure 1, please provide the size of B1 gene and each lane
Responses: revised accordingly. (line 246)
Point 9. Please revise line 285“revealed that 0.44 of the total……”
Responses: revised accordingly. (line 268)
Piont 10. The characters of Figure 5 are hard to read. Please provide font size that can be read.
Responses: We appreciate for Reviewer’s warm work earnestly and agree with this meaningful suggestion, the original Figure 5 has been revised.
Point 11. Line 403. “80%-90%” there should be a space between “ - ”.
Responses: revised accordingly. (line 387)
Piont 12. Lines 410, 410.·English Punctuations “,” should be used between phonologically proteins.
Responses: revised accordingly (lines 393-394).
Point 13. The references·format contain certain issues that require rectification, notably in references 2-15, and 43, among others. In addition, some proper nouns need italics, such as line 665, “in vivo”.
Responses: Thank you very much for your constructive suggestions. We have revised this accordingly.
We have done our best to address all comments and we sincerely hope that you find our MS revised to your satisfaction. We are looking forward to receiving your editorial decision soon.
With best wishes,
Jiyu Zhang, BVSc, MVSc, PhD
Key Laboratory of Veterinary Pharmaceutical Development,
Lanzhou Institute of Husbandry and Pharma-ceutical Sciences,
Chinese Academy of Agricultural Sciences, Ministry of Agriculture,
Lanzhou, Gansu Province 730050,
The People’s Republic China
Email: zhangjiyu@caas.cn
Jun-Jun He, PhD
College of Veterinary Medicine,
Yunnan Agricultural University,
Kunming, Yunnan Province 650201,
The People’s Republic China
Email: hejunjun617@163.com

Reviewer 2 Report
Comments and Suggestions for Authors
In this study authors utilized UHPLC combined with tandem mass spectrometer to analyze the phosphopeptides separated by LC-MS / MS. These techniques were performed to detect the post translational modification of proteins that plays a crucial role in cell signal transmission in Toxoplasma gondii infected host cells. The main goal of this investigations was to determine the changes in proteome of epithelial cells in the intestine of definitive host infected with Toxoplasma gondii.
To fulfil the aim of the study six cats were used in the experiment. Three of them were infected with Toxoplasma gondii Prugniuad (Pru) strain tissue cyst obtained from infected laboratory Kunming mice in which the strain was maintained. After 10 days animals were euthanized and samples were collected. For further molecular and proteomics investigations intestinal epithelial cells from small intestine of infected and non-infected cats were isolated.
The presence of Toxoplasma gondii infection in the cat intestinal epithelial tissues was confirmed using B1 gene PCR assay. To enrich labelling of phosphorylated peptides the TiO2 (Titanium dioxide) beads were used. Then iTRAQ labeling was performed. The phosphopeptides were separated by LC-MS / MS and analysed using UHPLC combined with tandem mass spectrometer. In this study 4,998 phosphopeptides were identified in collected samples. Using combined techniques of iTRAQ and TiO2 affinity chromatography analysis, 3,497 phosphorylation sites were identified within 1,805 phosphoproteins.
Among the differentially expressed 28 phosphoproteins, 68 were down-regulated and 637 were up-regulated. These significantly regulated phosphoproteins were further classified using GO analysis and KEGG enrichment. The analysis revealed that the infection with Toxoplasma gondii changed the properties of the phosphoproteins. It was connected with cellular compartments, including actin cytoskeleton reorganization and necroptosis of the host cell.
It was concluded that phosphoproteins play a role in the immune response of definitive host to infection with Toxoplasma gondii.
I find this study interesting and recommend it for publishing after minor corrections.
My suggestions:
Line 23: Toxoplasma gondii is a parasite that invades cells – what type of cells?
Line 24: This study aimed to understand how T. gondii regulates the definitive host cell signal transmission.
Line 25 and 26: The researchers … They used … correct form of the subject
Line 49: changes in the phosphorylation of proteins in the cat intestinal epithelia - intestinal epithelial cells
Line 105: the interaction between T. gondii and its definitive host – cells
Line: 126: intra gastric inoculation with 100 cysts / ml of sterile PBS – what is the infective dose? how many ml per cat? should be clarify.
Lines 128-130: Prior to collecting tissue samples, all animals were euthanized and the cat small intestinal epithelia were rinsed twice with PBS to ensure the removal of intestinal contents. At 10 days post-infection (DPI), the intestinal epithelial tissue of the cats was gently scratched using sterile techniques. – should be rewrite, change the order of sentences. “Ten days post infection …”
Line 133: then transported to BGI-shenzhen for sequencing using dry ice - ?
Line 137: Genomic DNA was extracted from each harvested tissues – what kind of tissue?
Line 173: The final washed pellets were … - ?
Lines 235 – 241: “All animals were handled strictly according to the Animal Ethics Procedures and Guidelines of the People’s Republic of China. The study was reviewed and approved by the Ethics Committee of Lanzhou Veterinary Research Institute (LVRI), Chinese Academy of Agricultural Sciences (CAAS) (Protocol Permit Number: LVRIAEC-2018-006). Efforts were made to minimize the suffering of cats and reduce the number of animals in the experiment.” – this paragraph should be moved
Line 244: After infecting the cat with Toxoplasma – when?
Line 253: (lanes 2-4, DPI_1, DPI_2, DPI_3) – should be rewrite – DPI_1 suggest 1 day post infection
Line 281: of cat small intestine infected by Toxoplasma gondii. – small intestine cells
Line 315: feline intestine infected by Toxoplasma gondii infection - ?
Line 360: Toxoplasma gondii is a wide-spreading pathogen
Line 366: T. gondii following host infection – host cells infection
Line 368: to investigate the phosphorylation process of the small intestine in the definitive host after T. gondii infection - ?
Line 388: in the small intestine of the definitive host following T. gondii infection – cells?
Line 411: the proliferation of T. gondii in the host – cells?
Comments on the Quality of English LanguageMinor editing of English language is required.
Author Response
9 Nov, 2022
Dr. Ewelina Bik
Assistant Editor for
Animals
Dear Dr. Ewelina Bik,
Re: Manuscript ID: animals-2632951.R1
On behalf of all co-authors, I would like to thank you and the reviewers very much for favorable comments and constructive suggestions on our manuscript (MS) animals-2632951. These comments and suggestions are very valuable for us to revise and improve the quality of our MS. We have revised the MS strictly according to the reviewers’ comments and suggestions. We used the “tracked changes” mode in the MS WORD to show the revised/changed text in the revised MS. Two MS files are uploaded: the “clean version” as “manuscript”, and the one showing “tracked changes” as “supplementary material”. In the following, we detail our point-by-point responses to the reviewer’s comments and suggestions. We made all our responses in blue colour for clarity.
Responses to the comments and suggestions of Reviewer #2
Comments and Suggestions for Authors
In this study authors utilized UHPLC combined with tandem mass spectrometer to analyze the phosphopeptides separated by LC-MS / MS. These techniques were performed to detect the post translational modification of proteins that plays a crucial role in cell signal transmission in Toxoplasma gondii infected host cells. The main goal of this investigations was to determine the changes in proteome of epithelial cells in the intestine of definitive host infected with Toxoplasma gondii.
To fulfil the aim of the study six cats were used in the experiment. Three of them were infected with Toxoplasma gondii Prugniuad (Pru) strain tissue cyst obtained from infected laboratory Kunming mice in which the strain was maintained. After 10 days animals were euthanized and samples were collected. For further molecular and proteomics investigations intestinal epithelial cells from small intestine of infected and non-infected cats were isolated.
The presence of Toxoplasma gondii infection in the cat intestinal epithelial tissues was confirmed using B1 gene PCR assay. To enrich labelling of phosphorylated peptides the TiO2 (Titanium dioxide) beads were used. Then iTRAQ labeling was performed. The phosphopeptides were separated by LC-MS / MS and analysed using UHPLC combined with tandem mass spectrometer. In this study 4,998 phosphopeptides were identified in collected samples. Using combined techniques of iTRAQ and TiO2 affinity chromatography analysis, 3,497 phosphorylation sites were identified within 1,805 phosphoproteins.
Among the differentially expressed 28 phosphoproteins, 68 were down-regulated and 637 were up-regulated. These significantly regulated phosphoproteins were further classified using GO analysis and KEGG enrichment. The analysis revealed that the infection with Toxoplasma gondii changed the properties of the phosphoproteins. It was connected with cellular compartments, including actin cytoskeleton reorganization and necroptosis of the host cell.
It was concluded that phosphoproteins play a role in the immune response of definitive host to infection with Toxoplasma gondii.
I find this study interesting and recommend it for publishing after minor corrections.
Responses: We thank Reviewer #2 very much for favorable comments and constructive suggestions on our MS.
More Comments:
Line 23: Toxoplasma gondii is a parasite that invades cells – what type of cells?
Responses: Toxoplasma gondii can invade a variety of nucleated cells, including humans and all warm-blooded animals. We thank Reviewer #2 very much for constructive suggestions. We have revised accordingly.
Line 24: This study aimed to understand how T. gondii regulates the definitive host cell signal transmission.
Responses: We thank Reviewer #2 very much for favorable comments and have revised accordingly.
Line 25 and 26: The researchers … They used … correct form of the subject
Responses: We thank Reviewer #2 very much for professional and helpful comments on our MS. We have revised accordingly.
Line 49: changes in the phosphorylation of proteins in the cat intestinal epithelia - intestinal epithelial cells
Responses: revised accordingly.
Line 105: the interaction between T. gondii and its definitive host – cells
Responses: revised accordingly.
Line: 126: intra gastric inoculation with 100 cysts / ml of sterile PBS – what is the infective dose? how many ml per cat? should be clarify.
Responses: 2 ml per cat. We revised this accordingly.
Lines 128-130: Prior to collecting tissue samples, all animals were euthanized and the cat small intestinal epithelia were rinsed twice with PBS to ensure the removal of intestinal contents. At 10 days post-infection (DPI), the intestinal epithelial tissue of the cats was gently scratched using sterile techniques. – should be rewrite, change the order of sentences. “Ten days post infection …”
Responses: We thank Reviewer #2 very much for professional and helpful comments on our MS. We have revised accordingly.
Line 133: then transported to BGI-shenzhen for sequencing using dry ice - ?
Responses: We are very sorry for our negligence of missing this in our MS. We thank the Reviewer #2 very much for helpful comments on our manuscript (MS). We have revised the manuscript accordingly “using dry ice transport”.
Line 137: Genomic DNA was extracted from each harvested tissues – what kind of tissue?
Responses: line 130 ‘A small portion of the collected samples (the intestinal epithelial tissue of the cats) was used for DNA extraction’ have already explained.
Line 173: The final washed pellets were … - ?
Responses: revised accordingly. (line 174)
Lines 235 – 241: “All animals were handled strictly according to the Animal Ethics Procedures and Guidelines of the People’s Republic of China. The study was reviewed and approved by the Ethics Committee of Lanzhou Veterinary Research Institute (LVRI), Chinese Academy of Agricultural Sciences (CAAS) (Protocol Permit Number: LVRIAEC-2018-006). Efforts were made to minimize the suffering of cats and reduce the number of animals in the experiment.” – this paragraph should be moved
Responses: removed accordingly.
Line 244: After infecting the cat with Toxoplasma – when?
Responses: we have added “at 10 DPI” (line 238).
Line 253: (lanes 2-4, DPI_1, DPI_2, DPI_3) – should be rewrite – DPI_1 suggest 1 day post infection
Responses: We thank Reviewer #2 very much for professional and helpful comments on our MS. We have revised accordingly (line 247).
Line 281: of cat small intestine infected by Toxoplasma gondii. – small intestine cells
Responses: revised accordingly (line 275).
Line 315: feline intestine infected by Toxoplasma gondii infection - ?
Responses: We thank Reviewer #2 very much for favorable comments on our MS, and we have revised accordingly (line 310).
Line 360: Toxoplasma gondii is a wide-spreading pathogen
Responses: revised accordingly. ‘line 354: Toxoplasma gondii is a wide-spreading intracellular parasite’.
Line 366: T. gondii following host infection – host cells infection
Responses: revised accordingly. (line 360)
Line 368: to investigate the phosphorylation process of the small intestine in the definitive host after T. gondii infection - ?
Responses: revised accordingly. (line 362)
Line 388: in the small intestine of the definitive host following T. gondii infection – cells?
Responses: revised accordingly. (line 382)
Line 411: the proliferation of T. gondii in the host – cells?
Responses: revised accordingly. (line 406)
We have done our best to address all comments and we sincerely hope that you find our MS revised to your satisfaction. We are looking forward to receiving your editorial decision soon.
With best wishes,
Jiyu Zhang, BVSc, MVSc, PhD
Key Laboratory of Veterinary Pharmaceutical Development,
Lanzhou Institute of Husbandry and Pharma-ceutical Sciences,
Chinese Academy of Agricultural Sciences, Ministry of Agriculture,
Lanzhou, Gansu Province 730050,
The People’s Republic China
Email: zhangjiyu@caas.cn
Jun-Jun He, PhD
College of Veterinary Medicine,
Yunnan Agricultural University,
Kunming, Yunnan Province 650201,
The People’s Republic China
Email: hejunjun617@163.com
